# Ti-Ions and/or Particles in Saliva Potentially Aggravate Dental Implant Corrosion

**DOI:** 10.3390/ma14195733

**Published:** 2021-10-01

**Authors:** Mostafa Alhamad, Valentim A. R. Barão, Cortino Sukotjo, Lyndon F. Cooper, Mathew T. Mathew

**Affiliations:** 1Department of Restorative Dentistry, College of Dentistry, University of Illinois at Chicago, Chicago, IL 60612, USA; malham7@uic.edu; 2Department of Restorative Dental Sciences, College of Dentistry, Imam Abdulrahman Bin Faisal University, Dammam 34212, Saudi Arabia; 3Department of Prosthodontics and Periodontology, Piracicaba Dental School, University of Campinas, Piracicaba, São Paulo 13414-903, Brazil; vbarao@unicamp.br; 4Department of Prosthodontics, School of Dental Medicine, Bahçeşehir University, Istanbul 34353, Turkey; 5Department of Oral Biology, College of Dentistry, University of Illinois at Chicago, Chicago, IL 60612, USA; cooperlf@uic.edu

**Keywords:** corrosion, titanium, dental alloys, peri-implantitis, dental implant

## Abstract

The corrosive titanium products in peri-implant tissues are a potential risk factor for peri-implantitis. There is very limited information available on the effect of the corrosion and wear products on the dental implant corrosion. Therefore, we determined the influence of Ti-ions and Ti-particles on Ti corrosion. Eighteen commercially pure-Ti-grade-2 discs were polished to mirror-shine. Samples were divided into six groups (n = 3) as a function of electrolytes; (A) Artificial saliva (AS), (B) AS with Ti-ions (the electrolyte from group A, after corrosion), (C) AS with Ti-particles 10 ppm (D) AS with Ti-particles 20 ppm, (E) AS with Ti-ions 10 ppm, and (F) AS with Ti-ions 20 ppm. Using Tafel’s method, corrosion potential (E_corr_) and current density (I_corr_) were estimated from potentiodynamic curves. Electrochemical Impedance Spectroscopy (EIS) data were used to construct Nyquist and Bode plots, and an equivalent electrical circuit was used to assess the corrosion kinetics. The corroded surfaces were examined through a 3D-white-light microscope and scanning electronic microscopy. The data demonstrated that the concentration of Ti-ions and corrosion rate (I_corr_) are strongly correlated (r = 0.997, *p* = 0.046). This study indicated that high Ti-ion concentration potentially aggravates corrosion. Under such a severe corrosion environment, there is a potential risk of increased implant associated adverse tissue reactions.

## 1. Introduction

Dental implant therapy is widely accepted in dental clinical practice owing to its high success rate. It is expected that 23% of adults in the United States could have dental implants by 2026 [1]. Titanium (Ti) is the most common biomaterial used to fabricate dental implants because of its high biocompatibility and inherent nature of protective film formation [2]. This oxide film has a thickness of 1.5–10 nm and is responsible for the superior corrosion resistance of this metal [3]. However, several factors can jeopardize the integrity of the titanium protective film, including surface friction and/or scratching during surgical implant placement or utilizing metal instruments for diagnosis or professional cleaning, excess cement, the galvanic effect created by potential dissimilarity of abutment and implant metals, titanium corrosion, and tribocorrosion due to the mastication [4,5,6,7].

A common cause of dental implant failure is peri-implantitis which is defined as the inflammation of the gingival tissues around a dental implant associated with concurrent bone loss [8]. The prevalence of peri-implantitis ranges between 10% and 30% [9]. Risk factors/indicators for peri-implantitis can be systemic such as uncontrolled diabetes and hormonal imbalance, or local such as a previous history of periodontitis, smoking, and plaque accumulation [8]. A research area of growing attention is that the presence of corrosive Ti products approximating peri-implant tissues could be a potential risk factor for peri-implantitis. In 2020, Soler et al. demonstrated an association between Ti corrosion and peri-implantitis by demonstrating the presence of Ti-particles in soft tissue biopsies collected from diseased implants [10]. However, whether Ti-particles/ions on the peri-implant tissues contribute to peri-implantitis is still debatable. In orthopedic literature, it is well established that wear debris is a major cause of osteolysis and aseptic loosening of the prosthesis [11]. In dentistry, this issue is still in question [12].

Several in vitro studies demonstrated the negative effects of Ti-particles/ions on cells including osteoblasts, fibroblasts, and macrophages. For example, in 2019, Schwarz et al. found that Ti-particles impaired the metabolic activity of osteoblasts and fibroblasts [13]. In 2017, Alrabeah and coworkers demonstrated an increase in bone resorption mediators, e.g., interleukin-6, interleukin-8, and receptor activator of nuclear factor kappa-B ligand (RANKL), in reaction to five different metal ions, including Ti [14]. Moreover, Ti-ions have the potential to cause a disbiotic condition by favoring the growth of periodontal pathogens [15].

The inflammatory response to Ti-ions and particles can be multifaceted. This does not only involve eliciting an acute inflammatory reaction [16], but can extend to chronic inflammation [17]. Furthermore, Ti-particles were found to negatively impact osteoblast viability and initiated programmed cell death when cultured with osteoblasts [18]. Ti-ions can have cytotoxic effects on osteoblasts, osteoclasts, and gingival epithelial cells [19]. These findings underscore the significance of investigating Ti corrosion and peri-implantitis association [14].

One of the missing links in understanding Ti corrosion is the effect of the presence of Ti products, i.e., Ti-particles and Ti-ions generated by tribocorrosion processes (mastication), in saliva on the further corrosion processes of the implant. To the best of our knowledge, there is only very limited published information available. Therefore, the purpose of this work was to determine the influence of Ti-particles and Ti-ions on the electrochemical kinetics of implant corrosion. Our first null hypothesis was that the presence of Ti-ions in artificial saliva does not increase the corrosion rate of the implant Ti surface. The second null *hypothesis* was that the presence of Ti-particles in artificial saliva does not increase the corrosion rate of the implant Ti surface

## 2. Materials and Methods

### 2.1. Sample Preparation

Eighteen commercially pure Ti grade 2 (as per the American Society for Testing Materials -ASTM, West Conshohocken, PA, USA) disks with a diameter of 11 mm and thickness of 7 mm were obtained by milling a Ti rod (Mac Master Carr, Elmhurst, IL, USA). The samples were divided into six groups (n = 3) as a function of electrolytes; (1) Artificial saliva (AS), (2) AS with Ti-ions (the electrolyte from group A, after the corrosion test), (3) AS with Ti-particles 10 ppm, (4) AS with Ti-particles 20 ppm, (5) AS with Ti-ions 10 ppm and (6) AS with Ti-ions 20 ppm. Prior to all tests, disks were polished to a mirror-shine and the polishing procedure involved the sequential use of grinding papers of 320-, 400-, 600-, 800- and 1200-grit (Carbimet 2, Buehler, Lake Bluff, IL, USA). The samples were cleaned in an ultrasonic bath after the use of each grinding paper. After that, a polishing cloth (TextMet Polishing Cloth, Buehler, Lake Bluff, IL, USA) impregnated with diamond paste (MetaDi 9-micron, Buehler) suspended in lubricant (MetaDi Fluid, Buehler) was utilized. To obtain a mirror-like shine (Ra values around 50 nm), the final polishing step involved the use of a polishing cloth (Chemomet I, Buehler) and colloidal silica polishing suspension (MasterMed, Buehler). Before all electrochemical tests, the samples were cleaned in an ultrasonic bath first in distilled water (10 min) followed by 70% propanol (10 min) and eventually dried with hot air (250 °C)

### 2.2. Electrolyte Preparation

The artificial saliva was prepared by adding the following components to distilled water at a physiologic pH of 6.5; KCl (0.4 g/L), NaCl (0.4 g/L), CaCl_2_·2H_2_O (0.906 g/L), NaH_2_PO_4_·2H_2_O (0.690 g/L), urea (1 g/L), and Na_2_S·9H_2_O (0.005 g/L). Each experiment was performed with an electrolyte volume of 10 mL. The electrolyte in group B was obtained from the control group solution following the electrochemical tests. The solution for groups C and D was prepared by adding Ti-particles (particle size >100 nm) (Sigma-Aldrich, St. Luis, MO, USA) to AS, resulting in a concentration of 10- and 20-ppm [18]. The solution for groups E and F was prepared by adding Ti Standard Solution-(NH_4_)_2_TiF_6_ in H_2_O (Merck KGaA, Darmstadt, Germany) to AS [15,19].

### 2.3. Electrochemical Tests

All electrochemical tests were done in a custom-made polysulfone cell (Figure 1). Electrochemical tests were performed with a three-electrode setting following ASTM guidelines (G-61). The reference electrode was a saturated calomel electrode (SCE), the counter-electrode was a graphite rod, and the working electrode was an exposed area (12.57 mm^2^) on the surface of the titanium sample. The electrochemical set-up was connected to a Potentiostat (G300, Gamry inc., Warminster, PA, USA) linked to a computer for data acquisition. A temperature of 37 °C was maintained throughout the experiment to simulate the oral environment. The corrosion experiment was initiated with an open circuit potential of 3600 s, followed by electrochemical impedance spectroscopy (EIS) measurements with a frequency range of 100 K to 0.001 Hz at open circuit potential (E_oc_). The samples were exposed to cyclic polarization with a potential range of −0.8 V to 1.8 V vs. SCE. Using Tafel’s estimation method, corrosion potential (E_corr_) and corresponding current density (I_corr_) were obtained from potentiodynamic curves. EIS data was used to construct Nyquist and Bode plots, and then an equivalent electrical circuit (modified Randle’s circuit) was developed to estimate solution resistance (R_s_), polarization resistance (R_p_) and double-layer capacitance (C_dl_).

### 2.4. Surface Analysis

To examine corroded and non-corroded surface areas, a white-light-interferometry microscope (Bruker–Nano Contour GT-K Optical Profilometer, Billerica, MA, USA) was used to obtain three-dimensional (3D) images. To quantify the surface changes of corroded and non-corroded areas, the surface roughness, Ra, was measured. To further check the surface quality, a scanning electronic microscope (Jeol JSM-IT500HR, Oxford Instruments, Oxford, UK) was used. Energy Dispersive Spectroscopy (EDS) spectra were obtained to analyze the surface elements.

### 2.5. Statistical Data Analysis

The statistical analysis was conducted for I_corr_, E_corr_, R_s_, R_p_, and C_dl_, in relation to control. At first, the normality test was performed for each variable. For data with normal distribution, one-way analysis of variance (ANOVA) was conducted to compare the impact of the electrolyte type on the corrosion processes, followed by Tukey’s post hoc analysis to detect the inter-group differences. The groups with non-normal distribution underwent Kruskal–Wallis test, a non-parametric statistical analysis. In this study, I_corr_, R_s_, R_p_, and C_dl_ were normally distributed, while E_corr_ was not normally distributed and followed a non-parametric test. All statistical tests were conducted using GraphPad Prism software at a significant level of 5%.

## 3. Results

### 3.1. Open Circuit Potential (OCP)

Representative graphs of the OCP are shown in Figure 2. In this study, the open circuit potential was used to indicate the corrosion tendency of the Ti sample (working electrode) compared to the reference electrode in the absence of any current applied to the corrosion system. For corrosion experiments, OCP of any given sample means that any potential applied below OCP indicates less corrosion tendency of the Ti sample, and any potential applied above OCP indicates more corrosion is expected to take place. Based on the representative OCP graphs, group C (Ti-particles 10 ppm) had the highest OCP potential while group E (Ti-ions 10 ppm) had the lowest OCP potential indicating more corrosion tendency in the latter.

### 3.2. Cyclic Polarization Curves

Representative potentiodynamic curves for each group are presented in Figure 3a, and the estimated corrosion parameters are presented in Figure 3b,c. The difference in electrochemical kinetics is displayed in potentiodynamic curves. The potentiodynamic curves show a transition from active to passive corrosion phases for all groups regardless of the electrolyte. Moreover, negative hysteresis followed the passive corrosion phase indicating the absence of pitting or crevice corrosion in all groups. Regarding I_corr_ and E_corr_, there was no statistically significant difference between any of the groups as compared to control (*p* > 0.05). However, data demonstrated that the concentration of Ti-ions and I_corr_ were strongly correlated with a correlation coefficient of 0.997 (*p* = 0.046). Conversely, the concentration of Ti-particles and I_corr_ values were negatively correlated with a coefficient of −0.997 (*p* = 0.049). The mean and standard deviation of E_corr_ and I_corr_ of the study groups are summarized in Table 1.

### 3.3. EIS Analysis

The obtained EIS data were studied through the Nyquist and Bode plots (Figure 4a,b). The Nyquist plot illustrates the electrochemical kinetics of the Ti surface. The Nyquist plots (Figure 4a) show that as the concentration of Ti-ions increased, it reduced the semi-circle of capacitance loop, indicating reduced corrosion resistance; on the other hand, the elevation of Ti-particle concentration increased the semi-circle of capacitance, indicating higher corrosion resistance.

Bode plots (Figure 4b) demonstrate the impedance at various frequencies of the electrochemical double bilayer developed at the area between the sample surface and the solution during the corrosion processes. Bode plots show that at high frequencies, the phase angle and impedance were low. At low frequencies, as the concentration of Ti-ions increased, the impedance and phase angle decreased, indicating higher corrosion tendency. In contrast, as the concentration of Ti-particles increased, the impedance and phase angle increased, indicating lower corrosion tendency.

To simplify the EIS data reading (EIS modeling), a simple equivalent circuit (modified Randle’s circuit with constant phase element (CPE)) consisting of solution resistance (R_s_), capacitance (C_dl_), and polarization resistance (R_p_) was utilized (Figure 5a). Using such an equivalent circuit, the electrochemical process taking place at the solution-titanium interface can be represented. In Figure 5a,b, the groups that have a different lowercase letter are statistically significant compared to control at *p* < 0.05. The results show a statistically significant decrease in R_s_ as the concentration of Ti-ions and Ti-particles increased (*p* < 0.05), indicating higher corrosion tendency except for group E. Furthermore, the data revealed that C_dl_ increased in Ti-ions groups B and E, which indicates lower corrosion resistance; however, this increase was not statistically significant (*p* > 0.05). Moreover, the presence of particles 20 ppm in the solution increased the polarization resistance R_p_ significantly (*p* < 0.05), indicating higher corrosion resistance.

### 3.4. Surface Characterization

The 3D white-light microscopy images show a difference in the surface topography between non-corroded and corroded areas (Figure 6a). The corroded areas have a significantly higher Ra value than non-corroded area in all groups confirming the occurrence of the corrosion processes (Figure 6b). In Figure 6b, different lowercase letters within each group indicate significant statistical difference (*p* ≤ 0.05). Additionally, SEM images displayed more surface irregularities on the corroded surfaces compared to non-corroded surfaces, especially in the ion groups (Figure 7a). On the SEM images of the ion groups, there is more visible roughness after corrosion compared to the other groups. EDS spectra demonstrated the presence of Ti, oxygen (O), carbon (C), and silicon (Si) (Figure 7b,c). The existence of C and Si on the sample surface could be due to surface contamination during the polishing processes.

## 4. Discussion

These simulation experiments demonstrated that the addition of Ti-ions to artificial saliva increased the corrosion processes of the Ti surface. Therefore, the first null hypothesis that the presence of Ti-ions in artificial saliva does not increase the corrosion rate of the implant Ti surface was rejected. The current study revealed that I_corr_ and the concentration of Ti-ions were directly proportional. R_s_ decreased significantly in the presence of Ti-ions 20 ppm, which may explain the increased corrosion as the solution conductivity increased. To the best of our knowledge, the effect of Ti-particles and Ti-ions in artificial saliva on corrosion has not been studied in detail. Previously, it has been reported that the Ti oxide layer was sensitive to a high concentration of electrolytes such as fluoride [7,20]. In 2006, Mabilleau et al. found that fluoride caused progressive damage to the Ti oxide layer as the concentration of fluoride increased [21]. However, it is worth noting that their study outcome may not be directly correlated with the current investigation (Ti-ions) due to the differences in the ionic nature.

The present study found that the presence of Ti-particles in the artificial saliva increased the corrosion resistance. This led to accepting the second null hypothesis. Ti-particle concentration in artificial saliva was inversely proportional to I_corr_. This unexpected outcome may have occurred for two possible reasons. First, the Ti-particles added to artificial saliva in this study are composed of titanium dioxide (TiO_2_), which possibly interfered with the electrochemical reactions due to TiO_2_ poor conductivity or insulating nature. In fact, TiO_2_ is used to produce coatings that increase corrosion resistance [22,23]. Second, the TiO_2_, used in this study, had a combination of rutile and anatase phases. In 2021, Pantaroto et al. found that rutile-anatase combination demonstrated higher corrosion resistance when used as a coating material [22]. However, several other factors might influence the increased corrosion resistance in the presence of Ti-particles, which requires further investigation.

In the current study, two concentrations, i.e., 10 ppm and 20 ppm, of Ti-ions and Ti-particles in artificial saliva were used. Similar concentrations were used to study the dose-effect of Ti-ions and Ti-particles on the biofilm growth. The study found that several pathogens related to the pathogenesis of peri-implantitis increased in elevated Ti-ion concentrations [15]. Another study looked at the effect of 0.5, 1.5, 5, 7.5, and 10 ppm of Ti-particles on osteoblasts and found that high concentrations, especially 10 ppm, negatively impacted the viability of osteoblasts [18]. In 2010, Mine et al. studied the impact of different Ti-ion concentrations on the viability of osteoblasts, osteoclasts, and gingival epithelial like-cells and found that exposure to Ti-ions 20 ppm significantly reduced the viability of those cells [19]. One of the factors that may lead to the release of Ti-products in the peri-implant tissues is tribocorrosion, which results from the synergistic effect of wear (due to mastication) and corrosion [24]. The released Ti particles could dissolve to form Ti-ions that accumulate in the peri-implant tissues, when present in body fluids such as the ginigival crevicular fluid or saliva [25].

EIS was used to describe the oxide layer behavior as it is considered a nondetrimental approach to corrode the metals [26]. In the present study, the representative Nyquist plots show that as the concentration of Ti-particles increased, the capacitance semi-circle increased, suggesting a thicker oxide layer. In contrast, as the concentration of Ti-ions increased, the capacitance semi-circle decreased indicating a declining corrosion resistance. On the representative Bode plots, particles 20 ppm increased both the impedance and the phase angle, suggesting lower corrosion tendency compared to particles 10 ppm. On the other hand, ions 20 ppm reduced the impedance and the phase angle compared to ions 10 ppm indicating higher corrosion tendency. Similar observations on Nyquist and Bode plots were found in a study performed by Barão et al. in which Ti corrosion was tested as a function of pH values [27]. Hence, there could be an effect of the chemical environment on the corrosion processes as a function of Ti-products, which is not in the scope of the current study.

From EIS modeling in the current investigation, R_s_ values suggest a higher corrosion resistance in the absence of Ti-ions and Ti-particles. As the EIS parameters (R_s_, R_p_ and C_dl_) indicate the local corrosion kinetics at the metal-solution interface, the decreasing R_s_ in the presence of Ti-ions suggests more susceptibility to corrosion, likewise decreasing Rp and increasing C_dl_. Despite showing a lower corrosion rate and higher corrosion resistance, both concentrations of Ti-particles reduced R_s_ significantly. In the Ti-ions groups, Ti-ions 20 ppm significantly dropped R_s_ while Ti-ions 10 ppm only caused a slight drop. Lower R_s_ indicates that the solution is more conductive and more likely to contribute to corrosion processes. The results of the present investigation revealed that the values of R_p_ and C_dl_ were statistically similar among the groups except for particles 20 ppm, which increased R_p_ significantly (*p* < 0.05). This perhaps originated from the slightly short duration of corrosion tests; more differences in R_p_ and C_dl_ may be clear across the groups at prolonged times [28]. Nevertheless, the particles 20 ppm group had the highest R_p_ and a lower C_dl_; which may indicate a higher protective effect of the oxide layer [27].

In the current study, three potential clinical conditions were simulated (Figure 8a). The first condition simulated Ti surface electrochemical behavior in contact with artificial saliva. In the second condition, two concentrations of Ti-ions, 10 ppm, and 20 ppm, were added to the artificial saliva. The third condition mimicked the presence of 10 ppm and 20 ppm of Ti-particles in artificial saliva. In the literature, the release of Ti-particles and ions on the peri-implant tissues during the surgical, prosthetic, and maintenance phases of dental implant treatment is well-established [29,30,31,32]. As previously mentioned in the current investigation, the concentration of Ti-ions was directly proportional to I_corr_; additionally, R_s_ significantly decreased when Ti-ions or Ti-particles were proximal to the Ti surface.

In the present study, the surface roughness of corroded areas was obtained and compared to the surface roughness of non-corroded areas using a 3D profilometer (Figure 6). It seems that the Ti-ions groups increased the surface roughness more than Ti-particles groups confirming the observations in the electrochemical tests. The samples were also scanned with SEM to further assess the surface quality (Figure 7). On the SEM images, more surface changes were observed in the Ti-ions groups reflecting ion’s effect on the corrosion processes. Higher surface roughness was found to favor biofilm formation [33]. Poor plaque control, therefore, increased plaque accumulation, was found to be strongly correlated to peri-implantitis [34,35]. In a real clinical scenario, the dental implant is subjected to a variety of complex conditions that may enhance the release of Ti-particles and ions in the peri-implant tissues. Examples for these conditions include pH variations, microbial changes, mechanical forces, and biocorrosion. However, it is uncertain how long the released Ti products are retained in the implant vicinity and whether they make an impact on the implant surface. Nevertheless, several studies reported the presence of higher Ti products on the tissues surrounding the implant, especially in case of peri-implantitis [2,36,37]. Moreover, in an in vitro study, Barão et al. demonstrated that LPS increased the Ti corrosion despite the fact that the duration of the standard electrochemical tests used is considered relatively short [28]. The current study used the same standard electrochemical tests and showed the negative impact of Ti-ions on the surface.

This study cannot account for cell-mediated effects that result from Ti-ion or particle exposure. Whether Ti-ions and Ti-particles are a risk factor for peri-implantitis is still debatable [8]. However, an in vitro study by Mine et al. demonstrated that Ti-ions significantly reduced the viability of osteoblasts, osteoclasts, and epithelial cells [19]. Furthermore, an in vitro study by Obando-Pereda et al. indicated that macrophages expressed more Toll-like receptors, which identify highly conserved pathogens and their elements, in the presence of Ti-particles [38]. The physical effects of Ti-ions/Ti-particles studied here may exacerbate potential cellular effects mediating peri-implant disease.

The results of the current investigation indicate that the presence of Ti-ions in artificial saliva increased the corrosion processes. Suggested is a Ti-ion-dependent positive feedback effect on the corrosion of implant Ti surface. In other words, high metal ion concentration in the vicinity of the implant potentially increases the risk of corrosion and, therefore, more peri-implant tissue irritation (Figure 8b). Although a direct causal relationship between peri-implantitis and Ti-ions as a risk factor is not well-established, a previous in vitro study investigated the pro-inflammatory impacts of different metal ions, including inflammasome stimulation, and found that Ti-ions aggregated to form Ti-particles that stimulated pro-inflammatory reaction [39]. The inflammatory reaction to Ti-particles and ions is not only limited to acute inflammation, but they can elicit chronic inflammation [17].

The major strength of the current study is that it is the first to provide evidence of the role of Ti products in the solution on the corrosion processes of Ti surface. These findings may direct future research projects in this area, as they highlight the increase of a potential risk factor for peri-implantitis. Additionally, the synergistic effect Ti products, inflammatory products and bacteria on Ti corrosion could be tested in future studies. Some studies in the orthopedic literature reported the occurrence of cell accelerated corrosion (CAC) with implant degradation, when the cells were challenged with metal particles, and identified the specific role of macrophages on such increased corrosion processes [40,41].

One of the limitations of this study is that Ti-particles used in the experiments may not represent the actual Ti-particles (size, chemistry, and shape) generated from different mechanical conditions surrounding the dental implant. The concern can be addressed to some extent by generating the Ti-particles from a dental implant simulator by applying the mastication forces, which we will consider in our future studies. Another limitation is that tests were performed on a polished surface while the dental implant surfaces can be with different surface roughness. Although the purpose of using a polished surface was to confirm the findings of the electrochemical tests and simplify the detection of any surface change, rougher implant surfaces may react differently to the corrosion processes. Furthermore, the tests conducted in this study only considered the electrochemical tests without the application of any mechanical forces (mastication), which may influence the corrosion processes. Further research that involves both corrosion tests and tribology (combined study is called tribocorrosion) should be conducted.

## 5. Conclusions

In this study, the influence of Ti-ions and Ti-particles in artificial saliva on the electrochemical behavior of Ti surface was investigated. The results indicated that the presence of Ti-ions in the artificial saliva in the vicinity of Ti surface may increase the corrosion processes. There was a strong correlation between the concentration of Ti-ions in artificial saliva and corrosion rate I_corr_. In contrast, Ti-particles decreased the corrosion of the Ti surface. The increased load of Ti products around a dental implant is a potential risk factor for peri-implantitis.

## Figures and Tables

**Figure 1 materials-14-05733-f001:**
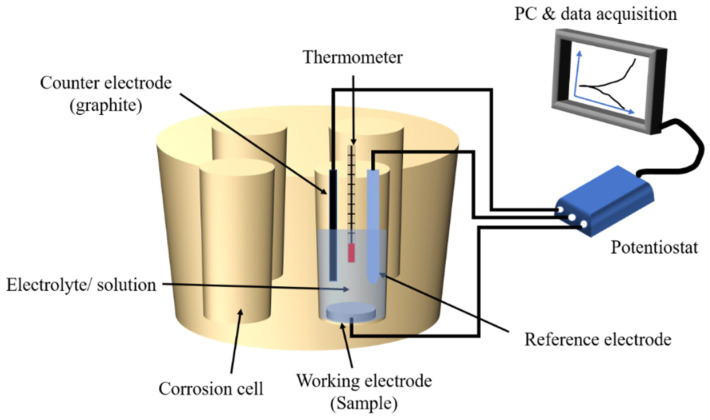
Schematic diagram of the electrochemical set-up.

**Figure 2 materials-14-05733-f002:**
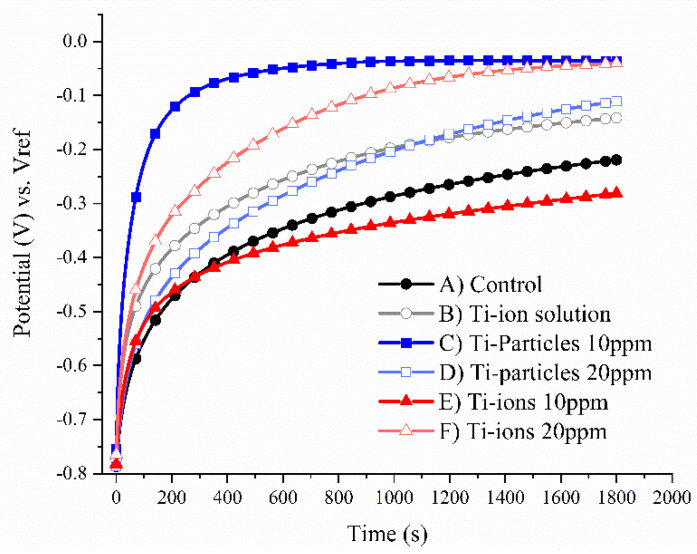
Representative open circuit potential (OCP) curves.

**Figure 3 materials-14-05733-f003:**
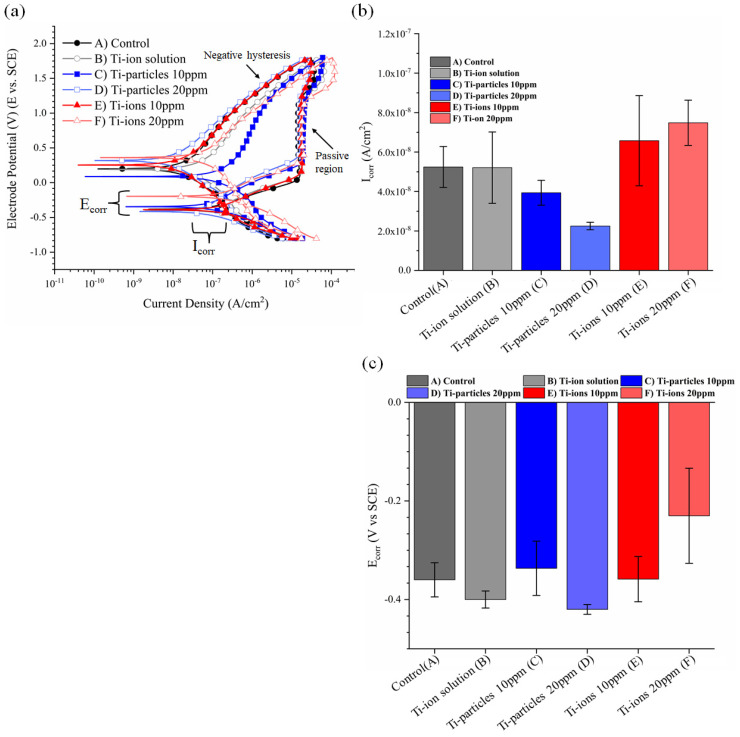
(**a**) Representative potentiodynamic curves from which the current density (I_corr_) and the corresponding potential (E_corr_) were obtained. (**b**) Mean and standard deviation of I_corr_ values of all groups. (**c**) Mean and standard deviation of E_corr_ of all groups.

**Figure 4 materials-14-05733-f004:**
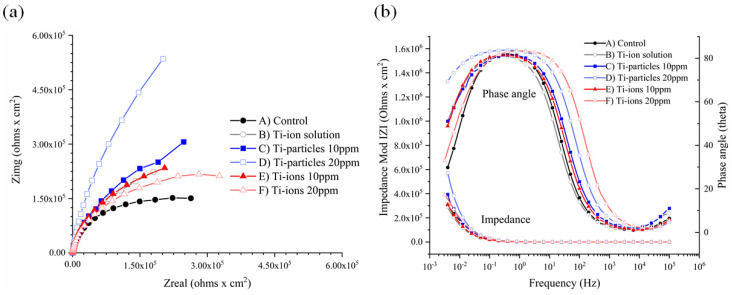
Representative (**a**) Nyquist plots and (**b**) Bode plots obtained from EIS tests with different electrolytes.

**Figure 5 materials-14-05733-f005:**
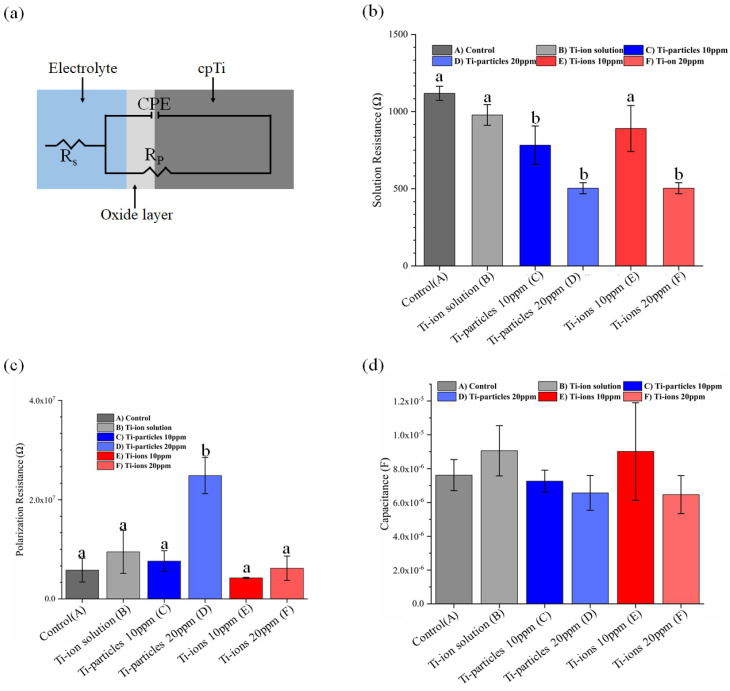
(**a**) Schematic diagram of modified Randle’s circuit. The change in (**b**) the solution resistance (R_s_), (**c**) polarization resistance (R_p_) and (**d**) double layer capacitance (C_dl_) for Ti surface in different Ti-ion and Ti-particle concentrations.

**Figure 6 materials-14-05733-f006:**
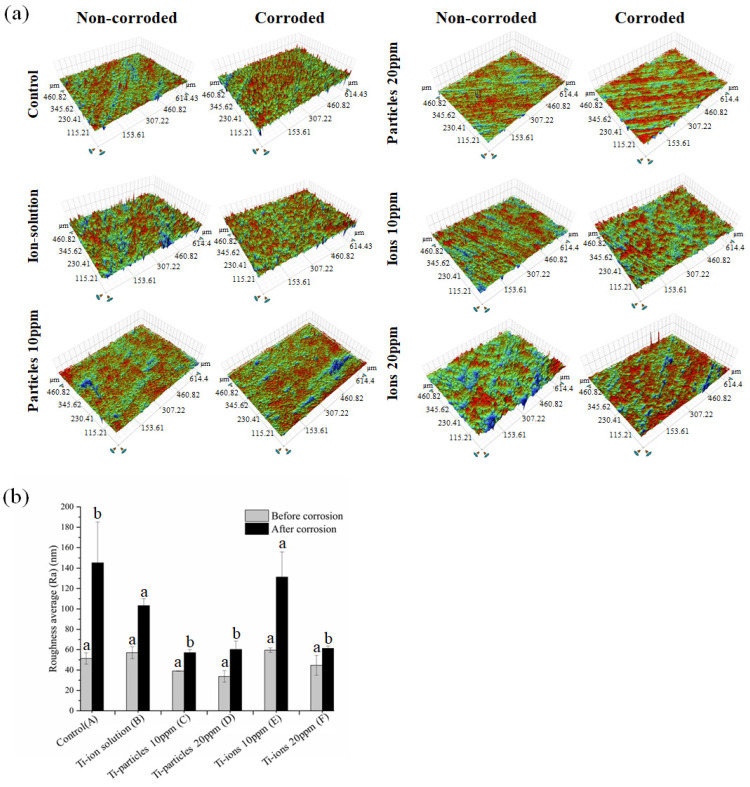
(**a**) White light 3D images of representative Ti samples. The non-corroded areas are shown in images in the left column and the corroded areas of the same samples are shown in the right column. (**b**) Mean and standard deviation of the surface roughness (Ra).

**Figure 7 materials-14-05733-f007:**
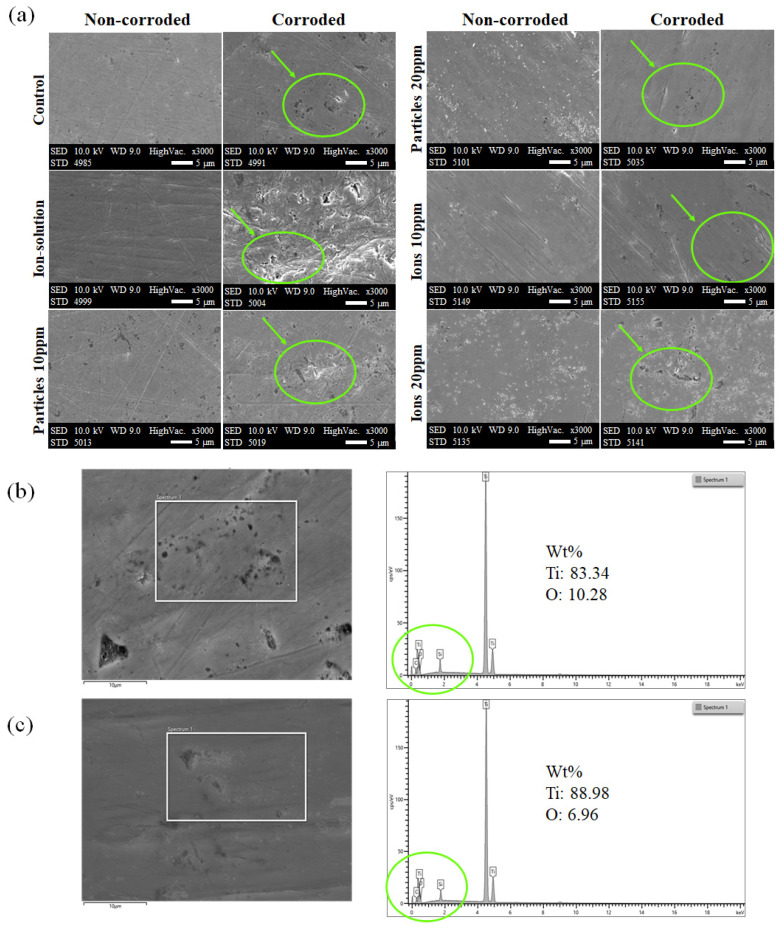
(**a**) SEM images of representative samples. The non-corroded areas are shown in the images in the left column and the corroded areas of the same samples are shown in the right column. The green circles and arrows indicate the surface irregularities caused by corrosion. EDS spectra of (**b**) control which show more oxygen indicating more passivation than (**c**) Ti-ion 20 ppm group.

**Figure 8 materials-14-05733-f008:**
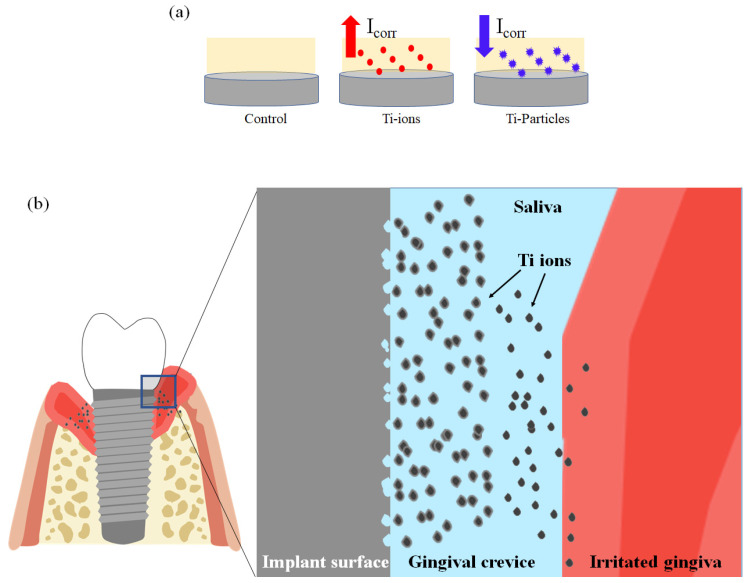
(**a**) The three major electrolytes used in this study and their relationship to I_corr_. (**b**) Schematic diagram illustrating the potential effect of Ti-ions in the vicinity of a dental implant and the surrounding tissue irritation.

**Table 1 materials-14-05733-t001:** Mean and standard deviation of I_corr_ (in A/cm^2^) and E_corr_ (in V) of the study groups.

Corrosion Parameter	Control	Ti-Ion Solution	Ti-Particles 10 ppm	Ti-Particles 20 ppm	Ti-Ions 10 ppm	Ti-Ions 20 ppm
I_corr_	5.24^−8^ (±1.04^−8^)	5.21^−8^ (±1.80^−8^)	3.94^−8^ (±6.29^−9^)	2.25^−8^ (±1.90^−9^)	6.58^−8^ (±2.28^−8^)	7.48^−8^ (±1.15^−8^)
E_corr_	−0.36 (±0.03)	−0.40 (±0.02)	−0.34 (±0.05)	−0.42 (±0.01)	−0.36 (±0.04)	−0.23 (±0.10)

## Data Availability

The data will be available upon request.

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
