# Peer review of "Ti-Ions and/or Particles in Saliva Potentially Aggravate Dental Implant Corrosion"

_materials, 2021, doi:10.3390/ma14195733_

Round 1

Reviewer 1 Report

This work investigated the effect of Ti-ions/particles in saliva on the corrosion of dental implants. It is important and useful. While, I have a few questions.

  1. Line 82, what is the size of the Ti particles and how did they suspend in the artificial saliva.
  2. Line 147, “while group F (Ti-ions 20ppm) had the lowest OCP potential”, isn’t it group E? Why?
  3. What is the meaning of Icorr and Ecorr shown in Figure 3 (a), it is obscure. What is the meaning of the error bars in Figure 3 (b) and (c)? Since the data always locates at the center of the error bar.
  4. Figure 5, what is the meaning and the significance of the statistically different? Could the authors explain more?
  5. Line 326 and 327, “Ti-ions in 326 the vicinity of Ti surface increased the corrosion processes”, while from the curves of Ti-ions 20 ppt in Figure 2, Figure 4 and Figure 6 (b), the conclusion is not so apparent.

Reviewer 2 Report

General remarks

The manuscript and the results of the investigation are of interest to the readers of the Materials. The subject of the research presented in the manuscript is the influence of Ti ions and Ti particles in artificial saliva on the electrochemical behavior of Ti surface, while the aim of the research was to determine the effect of Ti particles and Ti ions on the electrochemical corrosion kinetics of titanium implants. I agree with the authors that the issue of the influence of Ti ions and Ti particles in artificial saliva on periimplants is multidimensional. The most common cause of corrosion processes is the aggressiveness of the contact environment, which may initiate their processes. The process of destroying dental implants, which worsens the biofunctional properties of metal biomaterials, is usually the result of the biological metabolic activity of the body. These are mainly phenomena resulting from the reaction of hydrogen evolution and oxygen absorption, changes in body temperature, changes in saliva pH, operational factors - resulting from friction processes, mechanical damage, local overloads, incorrect implant geometry, the presence of specific ions or chemical compounds.

This discussion is based on two hypotheses: (1) the presence of Ti ions in the artificial saliva does not increase the corrosion of the Ti implant surface, and (2) the presence of Ti particles in the artificial saliva does not increase the corrosion of the Ti implant surface. Especially the first one is a significant simplification due to the failure to take into account the basics of knowledge in the field of thermodynamics and kinetics of electrochemical processes.

The scope of research on this multidimensional research issue has been limited to corrosion tests of a polished titanium surface (commercially pure Ti grade 2) covered with a native passive layer of titanium oxides in the environment of artificial saliva containing products of tribocorrosive processes, i.e. Ti-ions and Ti-particles under static conditions, at temperature of 37ºC. The study analyzed the results of open circuit potential (OCP), cyclic potentiodynamic curves, and electrochemical impedance spectroscopy (EIS) measurements. Estimated corrosion parameters, to their control statistical analysis was conducted, a schematic diagram of modified Randle's circuit was proposed. In addition, white-light-interferometry microscope was used to obtain three-dimensional (3D) images, surface of titanium topography and evaluation of corrosion damage symptoms. The research and the interpretation of the results do not raise any objections. Analyzes and their graphic presentation are careful.

The scope and area of ​​research was far from the actual operating conditions of titanium implants. However, the high susceptibility to the combined action of wear and corrosion (tribocorrosion) under the physiological environment is still a major concern. The release of Ti wear products along with the release of metal ions into the surrounding tissue and implant environment can lead to a huge number of side effects like inflammation, cytotoxicity, genotoxicity, carcinogenicity and in consequence implant failure. The comparison of the research results presented in this manuscript, their verification against the current state of knowledge of the results of tribocorrosion tests, according to the reviewer, should be presented in the discussion of the research results.

The manuscript requires the following corrections:

  1. The manuscript contains many editorial errors that should be eliminated, for example: the presence (in my opinion the concentration) of Ti ions was directly proportional to Icorr; peri-implantitis or periimplants? higher the corrosion; higher Ti-ions ;  increase corrosion of the implant Ti surface and other editorial errors.
  2. The subject of research and the analysis of the state of knowledge? The study concerned the corrosion processes in the environment containing tribocorrosive damage products. Thus, the state of the art should be presented that mainly relates to the tribocorrosive research results. Lack of a systematic approach to the analysis of the state of knowledge in the chapter Introduction and Discussion.
  3. The captions under the figures contain interpretations of the results of measurements / tests (they are repetitions) - transfer it to the text.
  4. Some of the statements are debatable, not very clear, e.g. concerning the influence of fluorine ions or the pH of the solution in the Discussion chapter.
  5. The statements contained in the paragraph - lines from number 214 to 220 are debatable, not very clear.

Conclusion:

This manuscript is interesting and delivers some valuable experimental results of the investigations. the article lacks a discussion (especially the corrosive properties / tribological properties) in the light of e.g. previous literature / tribocorrosive research results. In the opinion of the reviewer, in the case of an manuscript, a clear discussion is obligatory. Basing on the presented detailed remarks I am of an opinion that the evaluated work may be published in  the Materials, but before printing it needs major revision.

Reviewer 3 Report

The article "Ti-ions and/or Particles in Saliva Potentially Aggravate Dental Implant Corrosion" is well documented.

My comments and suggestion for improving it are as follows:

Abstract, line 23: "In Results," needs to be deleted.

Keywords, line 28: "and" has to be deleted.

Line 105: Please use Figure instead of Fig. Please check the entire manuscript, including Figure captions.

Line 106: American Society for Testing of Materials (ASTM) was previously defined in line 78. Please use the abbreviation only. 

Line 149. Figure 2 caption should be written using a proper font type and dimension. Same for next figures. "There is less thermodynamic tendency for group C with Ti-particles 10ppm to participate in the electrochemical corrosion. There is more thermodynamic tendency for group E with Ti-ions 10ppm to participate in corrosion processes." should be moved in the main text.

The results need to be summarized in one or several tables.

Authors contribution should be written in the proper format.

References should be written in the proper format.

Figure 3, caption: a. "were calculated." should be deleted. The explanation " For Icorr and Ecorr, although there is no statistically significant difference between the control and any of the groups at P <0.05, there is a strong correlation between Ti-ion concentration and Icorr (P < 0.05)" should be moved in the main text.

Same for Figure 4 caption: "On the Nyquist plot, as the Ti-ion concentration increases, the corrosion resistance decreases. On the Bode
plot, as the Ti-ion concentration increases, impedance and phase angle decrease indicating more corrosion tendency. ", Figure 5: "The groups that have a different lowercase letter are statistically different compared to control at P <0.05. For Cdl, there is no statistical difference between any of the groups and control." Figure 6: "The corrosion increased the surface roughness. Within each group, different lowercase letters indicate statistical significance (P≤ 0.05)"

Figure 7 caption: Scanning electron microscope (SEM) , energy dispersive
spectrometry (EDS). Please give full term only when first time used. These terms were previously used.
